# Experimental Evaluation of LoRaWAN Connectivity Reliability in Remote Rural Areas of Mozambique

**DOI:** 10.3390/s25196027

**Published:** 2025-10-01

**Authors:** Nelson José Chapungo, Octavian Postolache

**Affiliations:** Instituto de Telecomunicações—IUL (IT-IUL), ISCTE—Instituto Universitário de Lisboa, 1649-026 Lisbon, Portugal; opostolache@lx.it.pt

**Keywords:** LoRaWAN, wireless sensor networks, rural IoT, connectivity reliability, low-power communication, Mozambique

## Abstract

**Highlights:**

**What are the main findings?**
LoRaWAN demonstrated high packet delivery reliability (>89%) in a rural Mozambican environment with no electricity or mobile network.Signal strength and quality degraded significantly in lower altitudes, confirming the strong influence of terrain on connectivity.

**What is the implication of the main finding?**
Installing gateways on natural elevations improves LoRaWAN coverage in rural areas with uneven topography.LoRaWAN is a viable, low-cost, and scalable solution for smart agriculture and digital inclusion in disconnected regions.

**Abstract:**

This paper presents an experimental evaluation of the connectivity reliability of a LoRaWAN (Long Range Wide Area Network), deployed in a rural area of Mozambique, focusing on the influence of distance and relative altitude between end nodes and the gateway. The absence of telecommunications and power infrastructure in the study region provided a realistic and challenging scenario to assess LoRaWAN’s feasibility as a low-cost, low-power solution for remote sensing in disconnected environments. Field trials were conducted using an Arduino-based node (with 2 dBi antenna) powered by a 2200 mAh power bank, with no GPS or cellular support. Data were collected at four georeferenced points along a 1 km path, capturing Received Signal Strength Indicator (RSSI), Signal-to-Noise Ratio (SNR), and Packet Delivery Rate (PDR). Results confirmed that both distance and terrain elevation strongly affect performance, with significantly degraded metrics when the end nodes were located at lower altitudes relative to the gateway. Despite operational constraints, such as the need for manual firmware resets and lack of real-time monitoring, the network consistently achieved PDR above 89% and remained operational autonomously for over 24 h. The study highlights the effectiveness of installing gateways on natural elevations to improve coverage and demonstrates that even with basic hardware, LoRaWAN (Low Power Wide Area Network), is a viable and scalable option for rural connectivity. These findings offer valuable empirical evidence to promote national digital inclusion policies and future LPWAN deployments.

## 1. Introduction

The Internet of Things (IoT) has the potential to revolutionize sectors such as precision agriculture and environmental monitoring in low-infrastructure regions [1]. However, its effective implementation in rural areas of developing countries depends on technologies that combine long range, low power consumption, and affordable cost—core characteristics of LPWAN (Low Power Wide Area Networks) [2], as opposed to short-range networks such as Bluetooth, NFC, Wi-Fi/802.11, Z-Wave, and Zigbee [3], although they are also low-energy [4] consumption [5]. To further improve energy efficiency, a variant called BLE (Bluetooth Low Energy) was created in 2010, considered the best option for the new generation of IoT [6]. Among the LPWAN protocols, LoRaWAN stands out for its use of unlicensed spectrum, support for asynchronous communication, and minimal infrastructure requirements [7,8], ideal for the agricultural areas of Mozambique. Unlike Direct Sequence Spread Spectrum (DSSS) [9] and Frequency Hopping Spread Spectrum (FHSS) [10], LoRa utilizes Chirp Spread Spectrum (CSS) modulation, which offers greater resistance to interference and long-range transmission efficiency with low energy consumption. Its operating frequencies are 433, 868 and 915 MHz, which operate in ISM (Industrial, Scientific and Medical) bands, internationally reserved for unlicensed use [11].

According to the regional frequency allocation plan [12], in this study, the 868 MHz band was selected, as it is allocated by CRASA (Communications Regulators‘ Association of Southern Africa) [13], as an organization of SADC (Southern Africa Development Community), that promotes common regional policies for spectrum use.

Studies highlight the performance and limitations of LoRa in different contexts:Mining environments: A study conducted by Musonda, analyzed the challenges and design requirements to ensure the reliability of LoRaWAN in mining environments, characterized by underground structures and electromagnetic interference. The results highlighted the need for specific adjustments in the network configuration to maintain the integrity of transmitted data [14].Varied terrain: Research indicates that terrain topography significantly influences LoRa communication, potentially resulting in a reduction of up to 58.63% in signal reliability in areas with rugged terrain [15].Urban environments: Studies in densely populated urban environments have shown that despite interference, LoRaWAN can maintain reliability of up to 90.23% with appropriate configurations [16].Other study reinforces its effectiveness in remote environments, with ranges of up to 15 km in ideal conditions [17], while another one warns of trade-offs between latency and capacity in large-scale deployments—a relevant challenge for agricultural applications that require periodic transmissions, but not in real time [18].

In Mozambique, where 73% of the rural population does not have access to the internet [19] and electrical coverage is non-existent in vast areas [20], commercial solutions such as Starlink [21] or NB-IoT [22] are unviable due to prohibitive costs [23], continuous energy dependence [24] and centralized infrastructure. This reality reflects a broader scenario described by UN-HABITAT [25]: Rural areas in Africa often lack not only connectivity, but also basic sanitation and energy, perpetuating cycles of digital exclusion and inequality. As highlighted by Rosário [26], agribusiness in Mozambique is hindered by investment concentration in large-scale producers, leaving small farmers marginalized by digital technologies.

In this context, LoRaWAN emerges as a viable alternative, but its reliability in rugged terrains—typical of agricultural areas—still lacks local empirical validation [27]. Previous research has focused on urban or mining environments, neglecting critical rural conditions: topographic variations, dense vegetation, and moderate distances (1–5 km) with natural obstacles.

This gap is addressed through a field trial in an agricultural area of Mozambique, evaluating the impact of distance and relative altitude on metrics such as: RSSI (Received Signal Strength Indicator), SNR (Signal-to-Noise Ratio), PDR (Packet Delivery Rate), and Autonomy energetic.

Using low-cost equipment (Arduino Uno + LoRa Shield) and simulations of real conditions—without GPS, mobile network, or continuous power—our method replicates the constraints faced by smallholder farmers. Additionally, we tested practical strategies such as installing gateways on natural elevations, an approach aligned with the recommendations of the studies referenced above, but still little explored in African contexts.

The results demonstrate that, even with limited hardware, LoRaWAN achieved acceptable delivery rates over distances up to 1 km with autonomous operation.

This research offers:Empirical evidence for digital inclusion policies.Guidelines for implementation in similar contexts.Solutions aligned with local needs.

Although similar studies have assessed LoRaWAN in rural and agricultural settings globally, this research provides the first documented experimental evaluation in the African region completely disconnected from mobile and power infrastructure [28]. The unique combination of geographic isolation, topographic variability, and energy constraints addressed in this study offers novel empirical insights not covered by existing literature and contributes practical guidance for deployments under extreme infrastructural limitations.

### 1.1. Related Works

Recent literature has investigated LoRa and LoRaWAN performance under a variety of environmental and operational conditions. A specific propagation models for IoT network planning was proposed by a study in the Amazon region, showing that dense vegetation significantly affects path loss and must be considered for reliable deployment [29]. Other studies conducted preliminary field tests in woodlands, highlighting the impact of trees and foliage on RSSI and packet loss, particularly at distances above 200 m [30]. Similarly, vegetation loss effects in millimeter-wave bands were quantified, providing useful analogies about the attenuation introduced by natural obstacles, even though their focus was on higher frequencies [31]. More recently, a study evaluated LoRa propagation in maize fields, demonstrating how seasonal crop density and height directly influence connectivity, with implications for agricultural monitoring systems [32].

In addition to the recent works mentioned above, previous studies in Greece, Spain, and Ecuador have reported the feasibility of LoRa-based deployments in rural or disconnected areas, often under conditions of partial connectivity or institutional support [33,34,35]. In sub-Saharan Africa, only a few pilot projects in Kenya and Tanzania have explored LoRaWAN for applications such as irrigation monitoring in remote farms, demonstrating technical feasibility but still relying on partial infrastructure such as solar power or institutional support [36,37]. While these works confirm the robustness of LoRaWAN in diverse scenarios, most were conducted under controlled or partially connected conditions. In contrast, our study uniquely addresses a completely off-grid rural African environment, without mobile or power infrastructure, thereby filling a critical gap in the literature and complementing the limited African contributions with empirical evidence from Mozambique.

To contextualize the contribution of this study, Table 1 represents a comparative summary of six selected research works that evaluate LoRaWAN performance in different rural and disconnected scenarios (including this one). Diverse geographic settings are mentioned on these studies, they refer to Greece, Spain, Ecuador, Tanzania, Kenya, and Mozambique. While prior works demonstrate the viability of LoRaWAN in agricultural and infrastructure-limited environments, most were conducted in regions with partial connectivity, institutional support, or controlled pilot conditions. In contrast, the present study uniquely explores real deployment constraints in an off-grid Mozambican rural setting, characterized by the total absence of mobile coverage and energy infrastructure.

### 1.2. Contributions and Paper Structure

To the best of our knowledge, this is the first documented empirical evaluation of LoRaWAN connectivity in rural Mozambique under completely off-grid conditions, without mobile or power infrastructure. By addressing a region characterized by geographic isolation, topographic variability, and severe infrastructural limitations, this study complements and extends recent international works (Table 1) while filling a critical gap in the sub-Saharan African literature. The main contributions are threefold: (i) Providing empirical evidence of LoRaWAN performance in disconnected rural environments; (ii) deriving practical guidelines for IoT deployments in agricultural areas with no supporting infrastructure; and (iii) informing digital inclusion policies by demonstrating the viability of low-cost, energy-efficient technologies.

The remainder of this paper is structured as follows: Section 2 describes the experimental setup and methodology. Section 3 presents and analyzes the results. Section 4 discusses the findings in relation to prior work and highlights implications for practice and policy. Finally, Section 5 concludes the paper and outlines directions for future research.

## 2. Materials and Methods

### 2.1. Study Area and Operational Context

The study was conducted in a riverine agricultural zone in Nampula Province, northern Mozambique. This region is characterized by smallholder farms, absence of electricity, internet, and mobile coverage—conditions representative of rural Africa and suitable for testing LPWAN solutions.

The terrain includes modest natural hills, one of which was used to install the gateway, enhancing signal propagation without the need for towers. The test route (≈1 km) featured sparse vegetation, unpaved paths, and seasonal crops such as maize, cassava, rice, and horticultural species. Flooding during the rainy season and exposed river margins in the dry season shape cultivation patterns.

Although uninhabited, the fields are used by farmers who build temporary shelters, and scattered trees occasionally introduce minor signal obstructions. Figure 1 shows a ground-level view from point A, with the gateway on the hill and the final test point in the background, highlighting how natural elevations can serve as low-cost reception towers in disconnected environments.

### 2.2. Network Equipment and Architecture

The communication system was composed of a terminal node with Arduino Uno + LoRa Shield from Dragino with a 2 dBi antenna applied, a power bank, and Kora Mini Hub TBMH100 gateway connected to The Things Network (TTN).

The Dragino kit was selected for its affordability and compatibility with LPWAN protocols, making it viable for IoT prototyping [38]. It supports spreading factors from SF7 to SF12, offering a trade-off between transmission rate and range [39]. For this study, SF12 was configured to maximize range under irregular rural terrain conditions, consistent with technical recommendations for coverage-limited scenarios. Transmission power was set at 14 dBm, aligned with regulatory duty-cycle constraints for the EU868 band.

Table 2 summarizes the main technical characteristics of the end node. These parameters directly influence signal range, power consumption, and transmission reliability, making them essential for replicability and comparison with similar deployments. The configuration used in this study is consistent with technical recommendations and prior deployments of Arduino-based LoRa systems [40].

The Firmware development used the Arduino IDE and the LMIC library, widely applied in LoRaWAN applications. The system did not include GPS; instead, geolocation relied on Google Earth satellite imagery combined with direct physical observation. Both devices operated in the 868 MHz band, harmonized with CRASA allocations in the Southern African Development Community (SADC) [41].

Figure 2 illustrates the architecture of the communication system, including the data flow between the end node, gateway, and TTN server.

### 2.3. Experimental Procedure and Data Collection

Four measurement points (A–D) were selected along ~1 km, with altitude differences up to 30 m. The gateway remained fixed at point A (334 m).

In each point:The terminal node was manually restarted to trigger packet transmission.No real-time monitoring was possible due to lack of internet.Packets were later retrieved and analyzed on the TTN console.Metrics collected: RSSI, SNR, PDR, and node autonomy (battery duration).

Because of operational constraints (absence of automation and limited power), only 12 packets were transmitted per point, in compliance with the 868 MHz duty-cycle limit. Each manual restart triggered 2–4 packets, totaling 12 per location. Payloads were minimal to reduce time on air. Although this sample size limits statistical robustness, it provides a valid proof-of-concept for connectivity reliability under off-grid conditions. Future work will extend the dataset with automated packet generation and larger-scale tests.

The experiment lasted ~6 h under stable dry weather (≈32 °C, no rainfall). These conditions reflect typical irrigation demand periods; connectivity performance during rainy months will be addressed in future campaigns.

To complement the ground-level perspective previously shown in Figure 1, Figure 3 shows a georeferenced satellite view of the test area (Google Maps), marking the four points and route.

Geolocation of points:Point A: 15°14′01″ S, 39°25′10″ E.Point B: 15°14′05″ S, 39°25′26″ E.Point C: 15°14′07″ S, 39°25′28″ E.Point D: 15°14′08″ S, 39°25′31″ E.

### 2.4. Workflow of Experimental Design

To clarify the methodological process, a simplified workflow is provided:Configure end node (Arduino Uno + Dragino Shield).Select test point.Restart node manually (triggering 2–4 packets).Collect transmitted packets via TTN.Record RSSI, SNR, PDR.Repeat for all points.

A methodological diagram of this workflow is shown on Algorithm 1.
**Algorithm 1.** Experimental workflowInitialize end node (Arduino Uno + Dragino Shield, SF12, 868 MHz)FOR each test point in {A, B, C, D} DO   Place node at test location   FOR i = 1 to 3 restarts DO     Restart node     Transmit 2–4 packets     Log packet transmission time   END FOR   Retrieve packets from TTN console   Record RSSI, SNR, and PDREND FORAnalyze results with statistical tools (Excel, Python)

### 2.5. Data Presentation and Tools

Collected data were organized in tables and visualized through graphs. Microsoft Excel was used for initial tabulation, while Python libraries (matplotlib, seaborn, scipy) generated plots, regression lines, and Pearson correlation coefficients. This ensured both accuracy and reproducibility.

### 2.6. Limitations of the Experimental Design

The methodology presents several constraints:Use of a single end node and short distances (≤1 km).Limited packet sample (12 per point).Manual restarts required for packet generation.No real-time monitoring or automated GPS tracking.Measurements limited to dry-season conditions.

While these limitations reduce generalizability, they were intentional in this proof-of-concept study. They reflect the extreme operational constraints of rural Mozambique and demonstrate baseline feasibility. Future work will expand to multi-node setups, varied spreading factors, controlled payload lengths, and different environmental conditions.

## 3. Results

The experimental results demonstrate the direct impact of distance and altitude differences on LoRaWAN communication quality. Four georeferenced points (A–D) were analyzed along a 1 km transect, starting from the highest elevation (gateway) to the agricultural plain. Table 3 consolidates the main parameters recorded.

The following subsections present detailed analyses for each metric, supported by graphical representations.

### 3.1. RSSI

RSSI decreased consistently with increasing distance and decreasing altitude. The strongest signal was at point A (−14 dBm, 100 m, 334 m altitude), while the weakest was at point D (−108 dBm, 1000 m, 298 m altitude). Figure 4 shows the decline in RSSI across test points, reflecting attenuation due to distance and partial obstructions in line-of-sight conditions.

### 3.2. SNR

SNR followed a similar trend to RSSI. The best value (10 dB) occurred at point A, whereas point D registered the lowest (2.75 dB). Figure 5 demonstrates signal deterioration caused by terrain unevenness and vegetation obstacles.

### 3.3. PDR

Despite signal degradation, PDR values remained high. Point A achieved 100%, point B 97%, point C 91%, and point D 89%.

This robustness confirms LoRaWAN’s viability in harsh rural environments, consistent with findings from prior studies in agricultural and vegetated settings. Notably, a sharper PDR drop was observed beyond 500 m from the gateway. Figure 6 shows PDR as a function of distance.

### 3.4. Statistical Correlation Analysis

To strengthen the robustness of our findings, a Pearson correlation analysis was conducted between Packet Delivery Rate (PDR) and the two main signal quality indicators: RSSI and SNR. Strong positive correlations were observed (0.96 for PDR vs. RSSI and 0.97 for PDR vs. SNR), confirming that better signal quality directly improves packet delivery success in rural deployments. Additionally, the correlation between RSSI and SNR was weaker, indicating that while both metrics are related, they are independently influenced by terrain and propagation dynamics. These relationships are illustrated in Figure 7, which shows the pairwise correlations: (a) PDR vs. RSSI, (b) PDR vs. SNR, and (c) RSSI vs. SNR.

To enrich interpretation, future work will include larger datasets allowing computation of standard deviations, distributions, and boxplots for statistical robustness, as suggested by reviewers.

### 3.5. Comparison with Theoretical Path Loss Model

To validate the empirical measurements, the RSSI values obtained in the field were compared with predictions from a log-distance path loss model, using a reference distance of 1 m and a path loss exponent of *n* = 3, which is consistent with semi-rural terrain. Figure 7d presents this comparison. The measured RSSI values follow the same downward trend as the theoretical curve, remaining within a typical margin of deviation caused by environmental factors such as vegetation, soil humidity, and terrain irregularities. For example, at 1 km, the measured RSSI (−108 dBm) was only slightly below the predicted value, reflecting additional attenuation due to partial obstruction and reduced elevation.

This agreement between experimental data and theoretical expectations strengthens the reliability of the findings and confirms that the observed connectivity performance is representative of realistic deployment conditions in rural off-grid regions.

### 3.6. Comparison with Theoretical Propagation Model

To contextualize the empirical findings, the measured RSSI values were compared with a simple log-distance path loss model calibrated for 868 MHz. While the model predicts a smoother decay of signal strength with distance, the field data revealed sharper attenuation between 500 and 1000 m, likely due to terrain elevation differences and partial vegetation obstructions.

This comparison highlights the importance of conducting local field tests, as theoretical models alone may underestimate propagation challenges in African rural environments.

### 3.7. Box Plots for RSSI, SNR and PDR

In addition to the mean values reported in Table 3 and the correlation plots, Figure 8. Box plots of LoRaWAN performance metrics at the four test points: (a) RSSI, (b) SNR, and (c) PDR. The plots illustrate the distribution, variability, and median values across multiple transmissions at each point. These plots highlight the distribution and variability of the collected packets. While the overall trend confirms the degradation of signal quality with increasing distance and decreasing altitude, the box plots reveal slight fluctuations within each location, reflecting the impact of small-scale environmental dynamics such as vegetation density and minor obstructions. Despite this variability, PDR remained consistently high across all points, reinforcing the robustness of LoRaWAN for reliable communication even under constrained rural conditions.

### 3.8. Energy Autonomy

The terminal node operated continuously powered by a 2200 mAh power bank during the 6 h of testing. After this period, the device was kept on for more than 24 h with a full battery charge. This autonomy reinforces the low energy consumption characteristic of LoRaWAN and its suitability for environments without access to the electrical grid, as observed. Figure 9 shows LoRa device with 2 dBi antenna, connected to the power bank.

### 3.9. Related Work Comparison

To further validate the contribution, results were contrasted with recent studies in diverse environments, including dense vegetation, woodlands, maize fields, and rural agricultural pilots in Kenya and Tanzania.

A synthesis of these works was presented in Table 1 (Section 1.1. Related Works). While prior studies confirm LoRaWAN feasibility under partial connectivity or controlled conditions, the present results uniquely provide empirical evidence from a completely off-grid African agricultural setting, filling a critical gap in the literature.

## 4. Discussion

### 4.1. Effect of Relative Altitude and Distance

Even over short distances (<1 km), relative altitude was shown to strongly affect LoRaWAN performance. Points C and D, located 30 m below the gateway, presented the weakest indicators, reinforcing the importance of terrain in deployment strategies. Similar findings were reported in studies addressing vegetation and elevation impacts on LoRa propagation in the Amazon [29], woodland environments [30], and maize fields [32]. These works, along with our field evidence, emphasize that installation height and line-of-sight remain critical determinants of performance in rural deployments.

Placing gateways on naturally elevated areas (hills, rooftops, ridges) significantly increases effective coverage, representing a low-cost optimization strategy, as also noted in mining environments. In our case, this approach enabled reliable communication up to 1 km with minimal infrastructure, confirming its practical value for sub-Saharan Africa.

The strong correlations found between PDR, RSSI, and SNR are consistent with related literature [29,30,31,32], while also showing that rural Mozambican landscapes introduce unique challenges such as seasonal vegetation and scattered trees. Unlike controlled urban or industrial deployments, agricultural terrains are highly dynamic, requiring localized validation before large-scale adoption.

### 4.2. Comparison with Other Technologies

Short-range technologies (e.g., NFC, ZigBee, BLE) are inadequate for agricultural monitoring in disconnected rural areas due to their limited range. Long-range alternatives such as NB-IoT, Wi-SUN, and Starlink face additional constraints:NB-IoT: Requires mobile coverage and higher power consumption.Starlink: Relies on continuous electrical supply and incurs prohibitive costs [21].Wi-SUN: Promising for urban smart grids, but not widely available in remote African contexts.

By contrast, LoRaWAN offers low-energy consumption, ISM band operation, asynchronous communication, and compatibility with solar-based autonomous deployments. While Starlink may support rural schools or clinics with infrastructure, it does not address the economic conditions of smallholder farmers. For these, LoRaWAN emerges as the most cost-effective, sustainable solution.

### 4.3. Implications for Rural Projects

The energy performance observed aligns with existing literature on LoRa efficiency. In our setup, a 2200 mAh power bank sustained the device for more than 24 h, consistent with results in related agricultural deployments [29,30,31,32]. With solar integration, autonomy can be extended almost indefinitely, confirming LoRaWAN as suitable for:Off-grid operation in remote farms,Low-maintenance deployments, andIntegration with open platforms such as TTN.

Our results also showed that elevation relative to the gateway was decisive: points closer in altitude to the gateway consistently yielded higher delivery rates. This highlights the need for careful site planning in African lowland regions, prioritizing elevated spots (e.g., hilltops, cliffs, rooftops) for gateway installation.

Although this experiment used a single-node deployment, the results still achieved an average PDR of 94.25% across the four points. Literature shows that PDR can vary with higher node density and different payload sizes. Future work in our ongoing irrigation-focused IoT ecosystem will address these scenarios with multiple nodes and controlled configurations.

### 4.4. Limitations and Prospects for Improvement

The tests revealed several operational and technical constraints:No real-time monitoring due to lack of mobile network coverage,Absence of GPS, requiring manual georeferencing,Manual restarts for packet transmission, increasing variability,Use of a low-gain antenna (2 dBi), which limited the effective range to ~1 km.

While these limitations restrict scope, they also mirror realistic conditions for rural farmers without technical resources. Future iterations will address these gaps by:Deploying a small team to monitor connectivity during field trials,Optimizing firmware for scheduled automatic transmissions,Upgrading to LoRa-GPS modules to enable TTN Mapper integration,Using higher-gain antennas and improved elevation for extended coverage.

Some of these constraints may have introduced minor biases (e.g., variable restart times, approximate geolocation). However, they were intentionally accepted to simulate end-user limitations and inform more robust future deployments.

Ultimately, this proof-of-concept confirms LoRaWAN as a technically and economically feasible protocol for Mozambique’s rural digital inclusion, while outlining clear steps for scaling into multi-node, application-oriented IoT ecosystems.

### 4.5. Future Work

Future studies will address these limitations through:Deployment of multi-node testbeds with automated transmission scheduling to capture more robust statistics.Integration of GPS-enabled modules (e.g., LoRa GPS shields) to enable automatic georeferencing and mapping via TTN Mapper.Exploration of higher-gain antennas and elevated gateway installations to extend coverage in lowland agricultural areas.Incorporation of diverse weather conditions, especially rainy seasons, to evaluate performance under variable propagation environments.

In addition, this prototype forms part of a broader IoT ecosystem under development for smart irrigation in Mozambique. This next phase will involve multiple sensor nodes, water management integration, and controlled payload configurations, extending the empirical contributions of the present proof-of-concept to real-world agricultural applications.

## 5. Conclusions

This study demonstrated the feasibility of deploying LoRaWAN in rural Mozambican agricultural environments completely disconnected from electrical and mobile infrastructures. Field experiments showed that both distance and relative altitude significantly affect signal quality, with Packet Delivery Rates consistently above 89% and energy autonomy exceeding 24 h on a low-cost setup. The results confirm LoRaWAN’s efficiency for decentralized applications, highlighting the importance of gateway placement on elevated areas to mitigate coverage losses.

Despite operational constraints—manual restarts, lack of GPS, and limited real-time monitoring—the findings provide valuable empirical evidence under real-world rural conditions. Compared with alternative technologies such as NB-IoT or Starlink, LoRaWAN proved more suitable due to its use of unlicensed spectrum, low energy demand, and affordability, making it accessible to smallholder farmers.

This work contributes to the growing body of literature on IoT connectivity in Africa by addressing an extreme disconnection scenario rarely explored in prior studies. It offers practical guidelines for rural deployments and supports national strategies on digital agriculture and inclusion. Future efforts will extend this work through multi-node experiments, integration of GPS-enabled modules, automated transmissions, higher-gain antennas, and renewable energy sources. Ultimately, LoRaWAN stands out as a scalable and socially impactful solution for smart irrigation, crop monitoring, and rural alert systems, aligning with initiatives such as Mozambique’s Agriculture Digital Strategy and the global “Partner2Connect” agenda.

## Figures and Tables

**Figure 1 sensors-25-06027-f001:**
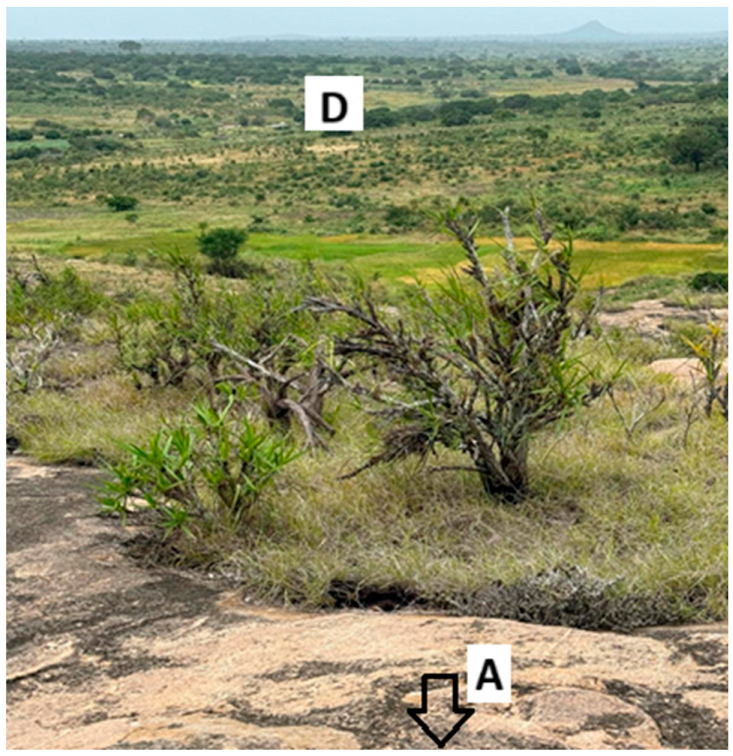
Line vision—field view of the study area showing the terrain profile, seasonal crops. The arrow indicates the approximate location of the gateway (not visible in the image). Letters A and D mark the initial and final test points along the route.

**Figure 2 sensors-25-06027-f002:**
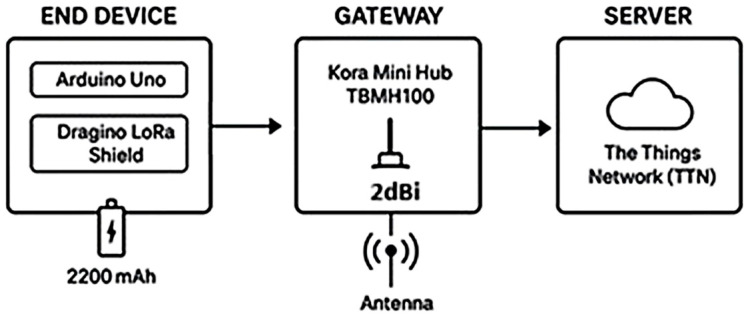
Architecture of the LoRaWAN communication system used in the field.

**Figure 3 sensors-25-06027-f003:**
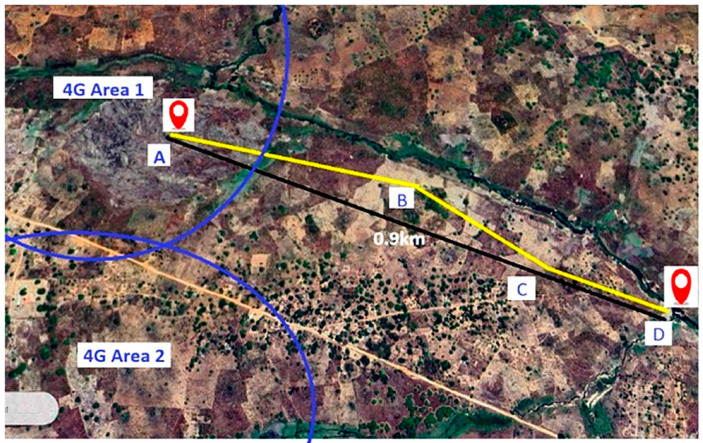
Georeferenced map showing the 0.9 km with the trajectory from point A to D indicated by yellow line, the black line shows the straight-line distance, and the blue arcs represent the two 4G coverage areas (4G area 1 cover the point A, where the gateway is installed).

**Figure 4 sensors-25-06027-f004:**
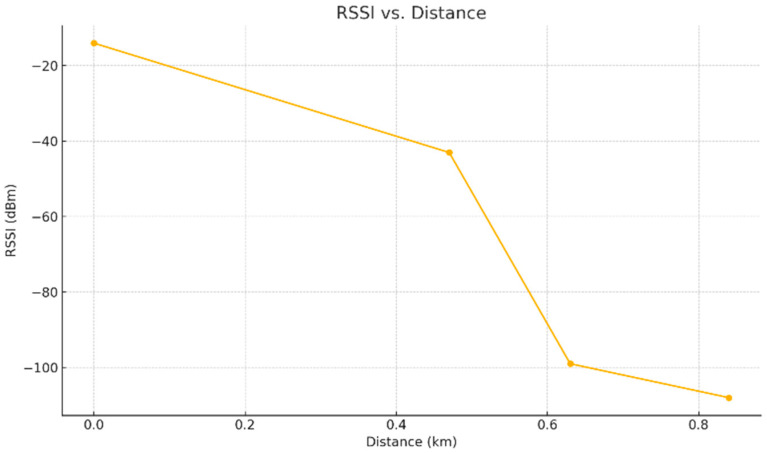
RSSI per measurement point.

**Figure 5 sensors-25-06027-f005:**
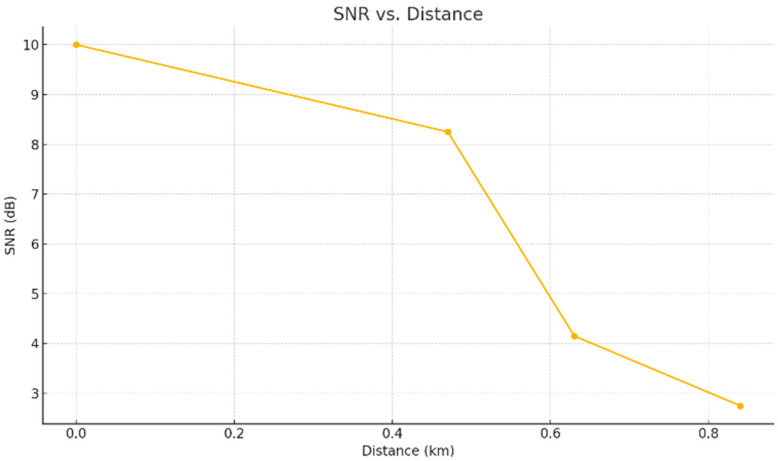
SNR by measurement point.

**Figure 6 sensors-25-06027-f006:**
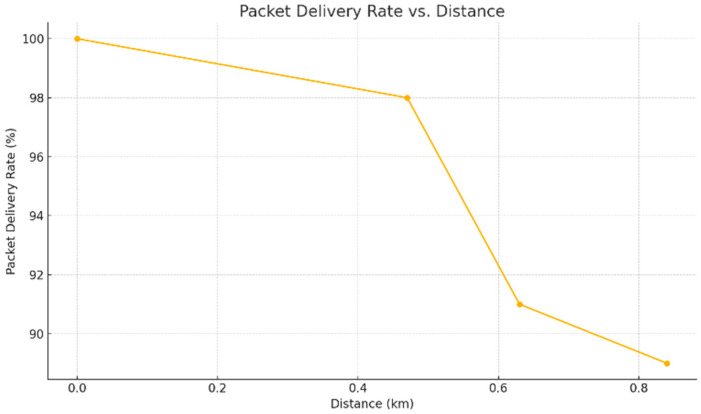
PDR variation as a function of distance.

**Figure 7 sensors-25-06027-f007:**
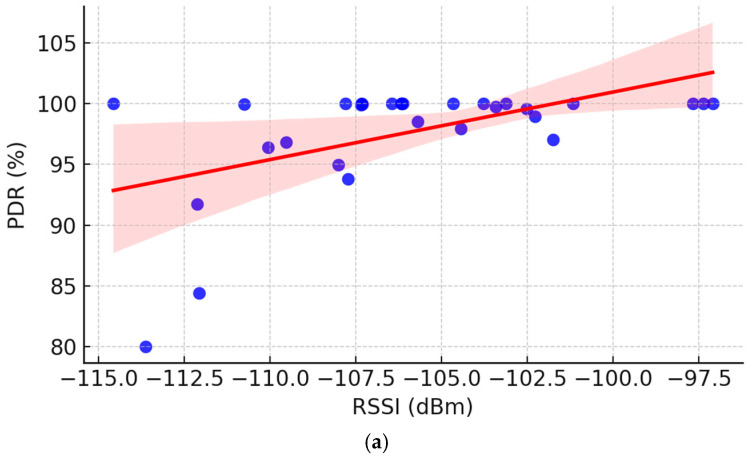
(**a**) Correlation PDR vs. RSSI: Stronger signal strength improves delivery success. (**b**) Correlation PDR vs. SNR, highlighting that higher SNR values favor packet delivery in rural deployments. (**c**) Correlation between RSSI and SNR, revealing a weaker alignment due to distinct environmental propagation effects. (**d**) Comparison of experimental RSSI with log-distance path loss model: Measured RSSI follows the general trend of the log-distance model (n = 3.0) but shows significant scatter due to environmental factors, the orange solid line represents the theoretical log-distance, predicting signal attenuation under idealized conditions. The blue dashed line connects the actual RSSI measurement. In (**a**–**c**), blue dots are individual measurements, the solid red line is the fitted trend, and the pink band represents its 95% confidence interval.

**Figure 8 sensors-25-06027-f008:**
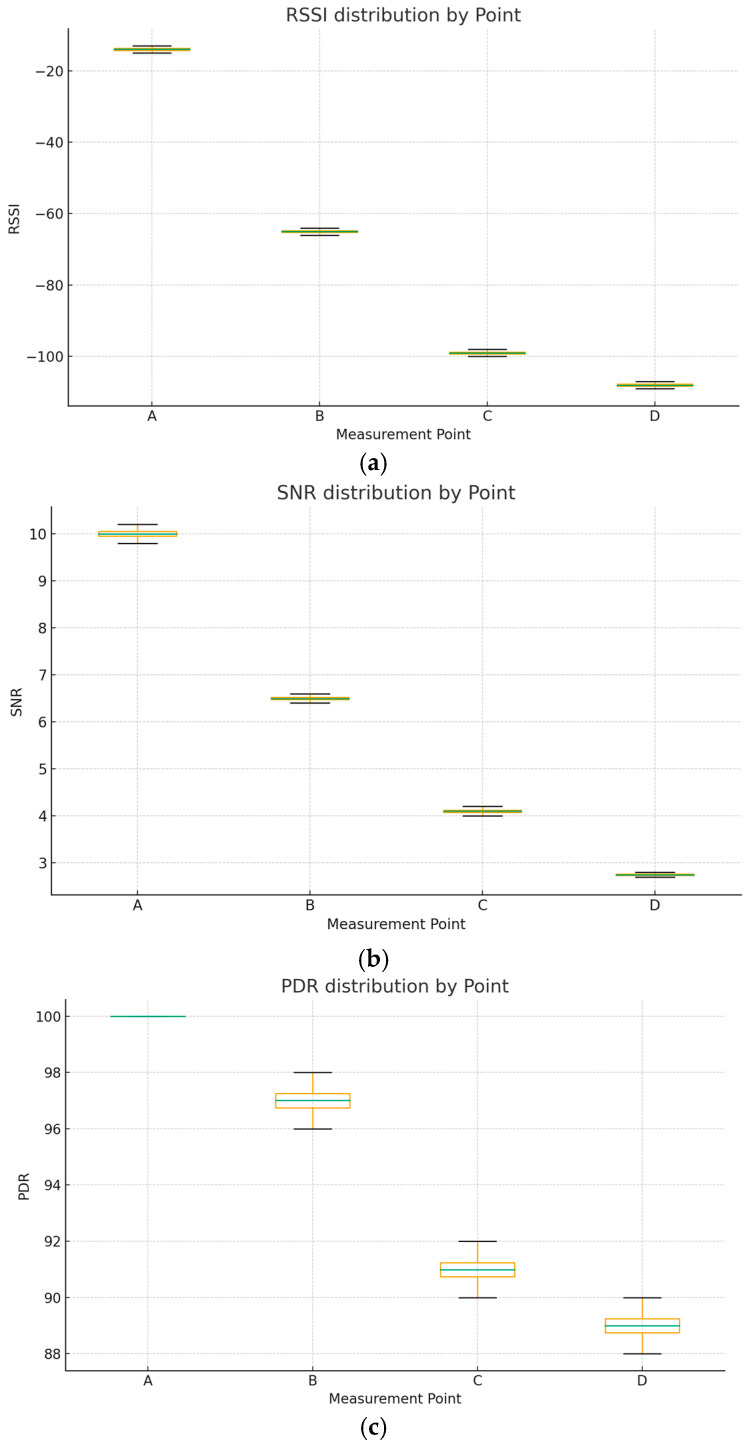
(**a**) RSSI by measurement point (box plot): Distribution of RSSI values across all test points, showing progressive degradation with distance. (**b**) SNR by measurement point (box plot): Distribution of SNR values, highlighting the reduction in signal quality at lower altitudes. (**c**) PDR by measurement point (box plot): Distribution of PDR values, confirming high delivery success rates with mild variability across points.

**Figure 9 sensors-25-06027-f009:**
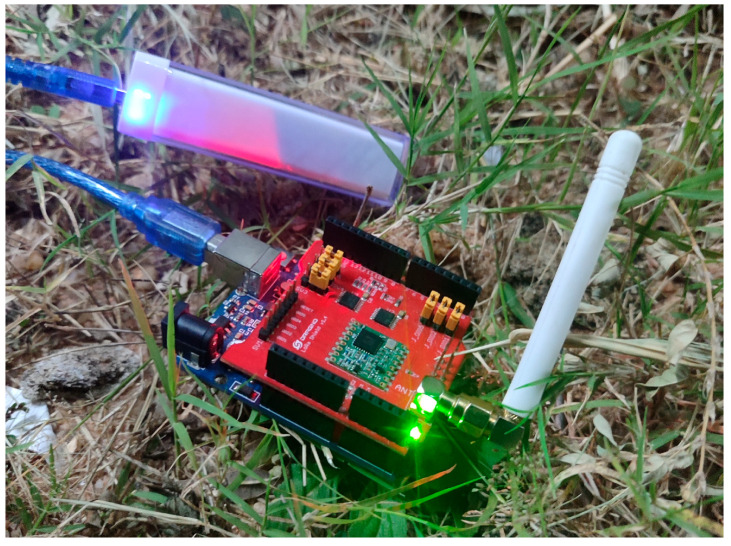
Terminal device: Arduino Uno + LoRa Shield and power bank power supply.

**Table 1 sensors-25-06027-t001:** Related work comparison.

Location	Scenario	Technology	Key Findings	Limitations
Greece	Urban and rural	LoRaWAN	High PDR in rural areas; tested coverage compared to urban scenarios.	Structured terrain and presence of GSM infrastructure.
Spain	GSM dead zones	LoRa Mesh	Demonstrated feasibility of LoRa in blind spots.	Indoor-focused pilot; limited scalability evaluation.
Ecuador	Agricultural field	LoRa, ZigBee	LoRa showed best trade-off for energy efficiency and distance.	Mild terrain; GPS and automation support used.
Tanzania	Rural agriculture	LoRaWAN	Real-time soil monitoring is feasible using LoRaWAN.	Deployment limited to school-owned farms with some infrastructure support.
Kenya	Irrigation control	LoRaWAN	Improved irrigation efficiency and water access with LoRaWAN.	Powered by solar panels (not fully autonomous).
Amazon, Brazil	Dense vegetation	LoRaWAN	Developed propagation models; vegetation strongly impacts path loss.	Focused on 915 MHz; results may differ in 868 MHz environments.
Woodlands	Vegetated areas	LoRaWAN	Preliminary tests show foliage causes notable RSSI reduction and packet loss.	Short distances (<200 m); limited test scenarios.
Poland	Single alley trees	mmWave bands	Measured vegetation-induced signal loss; useful analogies for wireless IoT.	Different frequency band; indirect applicability to LoRaWAN.
China	Maize fields	LoRaWAN	Seasonal crop height and density directly affect propagation and reliability.	Study limited to controlled crop fields; does not replicate off-grid context.
Mozambique (this study)	Remote, off-grid agriculture	LoRaWAN OTAA	Achieved >89% PDR using low-cost hardware in absence of GSM and power.	Manual operation and single-node setup; short-range (1 km).

**Table 2 sensors-25-06027-t002:** Technical specifications of the end node.

Parameter	Value
Microcontroller	Arduino Uno (ATmega328P, DigiKey, Lisbon, Portugal) powered by a 2200 mAh battery
LoRa Shield Model	Dragino LoRa Shield (SX1276, DigiKey, Lisbon, Portugal) with a 2 dBi antenna
Frequency Band	868 MHz ISM band
Spreading Factor (SF)	SF7 to SF12 (configured SF12)
Bandwidth	125 kHz
Transmission Power	Up to 20 dBm (set to 14 dBm)
Activation Method	OTAA (Over-The-Air Activation)
Duty Cycle	1% (duty cycle limit for EU868)

**Table 3 sensors-25-06027-t003:** Indicators by measurement point.

Point	Coordinates	Distance (m)	Altitude (m)	RSSI (dBm)	SNR (dB)	PDR (%)
A	15°14′01″ S, 39°25′10″ E	100	334	−14	10	100
B	15°14′05″ S, 39°25′26″ E	500	322	−65	6.5	97
C	15°14′07″ S, 39°25′28″ E	750	300	−99	4.1	91
D	15°14′08″ S, 39°25′31″ E	1000	298	−108	2.75	89

## Data Availability

Data supporting the findings of this study are contained within the article. Additional details may be provided upon reasonable request from the corresponding author.

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
