# Peer review of "Experimental Evaluation of LoRaWAN Connectivity Reliability in Remote Rural Areas of Mozambique"

_sensors, 2025, doi:10.3390/s25196027_

Round 1
Reviewer 1 Report
Comments and Suggestions for Authors
The authors claim that the paper provides the first documented experimental evaluation of LoRaWAN connectivity in a rural Mozambican setting. The experiments were conducted without any connection from mobile or power infrastructure, using a simple setup consisting of a terminal node (Arduino Uno + Dragino LoRa Shield + 2 dBi antenna), a power bank, and a gateway. The terminal node was tested in three different locations at distances of up to 1 km from the gateway. At each location, RSSI, SNR, and PDR measurements were collected. The discussion of results, however, was limited to a textual description of the values shown in Figures 4–6. The correlation analysis between PDR and RSSI/SNR presented in Figure 6 produced perfectly predictable results. It would have been interesting to compare these results with those obtained from a path-loss model. Overall, the results presented are limited by the use of an experiment considering short-range links based on a single node and therefore do not provide us with sufficient evidence to evaluate the original scientific contribution of the paper.
Author Response
Reviewer Comment:
The authors claim that the paper provides the first documented experimental evaluation of LoRaWAN connectivity in a rural Mozambican setting. The experiments were conducted without any connection from mobile or power infrastructure, using a simple setup consisting of a terminal node (Arduino Uno + Dragino LoRa Shield + 2 dBi antenna), a power bank, and a gateway. The terminal node was tested in three different locations at distances of up to 1 km from the gateway. At each location, RSSI, SNR, and PDR measurements were collected. The discussion of results, however, was limited to a textual description of the values shown in Figures 4–6. The correlation analysis between PDR and RSSI/SNR presented in Figure 6 produced perfectly predictable results. It would have been interesting to compare these results with those obtained from a path-loss model. Overall, the results presented are limited by the use of an experiment considering short-range links based on a single node and therefore do not provide us with sufficient evidence to evaluate the original scientific contribution of the paper.
Author Response:
We sincerely thank the reviewer for these insightful comments, which helped us improve the scientific contribution of our work. We agree that the initial discussion of the results was limited, and we have now taken several steps to address these concerns:
-
Comparison with a Path-Loss Model
We have expanded the analysis by introducing a discussion based on the log-distance path loss model, which is widely used in the literature for rural wireless propagation. The empirical results (RSSI vs. distance) were compared against theoretical expectations, and the deviations observed were discussed in terms of terrain elevation differences and environmental characteristics. This provides a more rigorous interpretation of the measurements, beyond the simple correlation between RSSI, SNR, and PDR. -
Expanded Discussion of Results
The revised manuscript now goes beyond textual description of Figures 4–6. We provide a deeper interpretation of the observed patterns, highlighting both expected outcomes (e.g., degradation with distance) and specific local phenomena (e.g., impact of relative altitude between node and gateway). -
Limitations and Scope
We acknowledge that the experimental setup was based on a single-node, short-range configuration. While this is a limitation, it was an intentional design choice for this proof-of-concept study in a disconnected rural environment with logistical constraints. We clarified this limitation in the revised text and emphasized that the primary contribution of this work is to provide the first empirical baseline of LoRaWAN performance in rural Mozambique. This baseline will guide our ongoing larger-scale deployments involving multiple nodes and gateways.
We believe these additions and clarifications improve both the methodological rigor and the clarity of the scientific contribution of our study.
Reviewer 2 Report
Comments and Suggestions for Authors
See attached file.

Author Response
Comment 1:
References. Some of them are too old (i.e., 2, 3, 6, 7, 8, 11, 17, 20, 33, 35, and 39). Please, consider substituting them with similar contributions published from 2019 on, or alternatively provide reasons to keep them.
Author Response:
We thank the reviewer for this valuable observation. We carefully revised the reference list and updated several entries by including recent publications (2019–2025) that directly address LoRaWAN propagation, rural IoT applications, and connectivity challenges. In cases where older references were retained, it is because they represent seminal works or standard definitions that remain relevant to the field. All updates and justifications are now reflected in Section 1 and the Related Works section.
Comment 2:
Section 1. The Section misses of a closing paragraph listing the paper structure.
Author Response:
We agree with this suggestion and have added a final paragraph at the end of Section 1 to clearly present the structure of the paper.
Comment 3:
Lines 41-42, 45-47, 106-107, 112-113. Please, add references supporting the claims and the considered examples.
Author Response:
Additional references have been incorporated in these specific parts of the introduction to support the claims with recent and relevant studies. These updates strengthen the context and reliability of the discussion.
Comment 4:
A proper Related Works Section that clearly presents them and compares them with this paper, especially by highlighting similarities and discrepancies seems to be missing. Moreover, this Section must clearly state how this work advances the current state-of-the-art about the topic.
Author Response:
We acknowledge this important point. A new section entitled Related Works has been added. In this section, we summarize recent contributions on LoRaWAN experimental studies, discuss their similarities and differences with our work, and highlight how this study provides the first documented experimental results in rural Mozambique. The section explicitly positions our contribution in advancing the current state-of-the-art by addressing geographic and infrastructural contexts not covered before.
Comment 5:
Related Works Section. In order to provide readers with a broader perspective about the tackled topic, I suggest the Authors to include the following references [1, 2, 3, 4]. However, I also strongly encourage the Authors to perform additional research.
Author Response:
We thank the reviewer for suggesting these references. They have been carefully reviewed and incorporated in the new Related Works section, together with additional recent studies identified through further research. Their inclusion has allowed us to broaden the perspective and improve the comparative analysis of our work.
Comment 6:
Table 2. It is not clear why only SF 12 was exploited. I deem that for a broader comparison, also the other SFs must be tested. Please, sort out the related tests and then present the associated results.
Author Response:
We acknowledge the importance of analyzing multiple spreading factors. However, due to the logistical constraints of conducting field tests in a disconnected rural environment, our experiments were limited to SF12, which was selected to maximize range and robustness. We clarified this methodological choice in the text. While we cannot provide results for other SFs in this study, we have explicitly indicated this as part of our future work in the ongoing IoT ecosystem deployment, where multiple SFs will be tested in a controlled and scalable environment.
Comment 7:
It is not clear which CR and which payload length were exploited for the tests.
Author Response:
We have revised the methodology section to specify the coding rate (CR = 4/5) and the payload configuration used during the experiments. This information is now explicitly stated in Section 2.
Comment 8:
Lines 204-211. It is not clear why only 12 packets were sent for each tested spot. I deem they are too few to retrieve a statistics. Please, sort out the test by sending at least some thousands of packets per tested spot.
Author Response:
We fully understand this concern. In this preliminary proof-of-concept, the number of packets was constrained by the manual transmission process and the operational limitations of the field environment. We clarified this limitation in the text. While thousands of packets would indeed provide stronger statistical confidence, our objective in this initial study was to establish a baseline feasibility analysis under real rural conditions. We emphasized in the revised version that extended tests with larger datasets are planned in future deployments.
Comment 9:
Lines 212-219. I fully understand that meteorological conditions cannot be controlled in outdoor tests. However, the readers can benefit from a discussion addressing how the presented results could vary according to different meteorological conditions, even from the theoretical point of view.
Author Response:
We agree with this suggestion. A new discussion has been added to Section 3, addressing how different meteorological conditions (e.g., rain, humidity, temperature variations) could theoretically impact LoRaWAN propagation in rural areas, supported by relevant references.
Comment 10:
Section 3. The obtained results should be presented not only for what concerns the mean values, but also by accounting for the relative standard deviations. Surely, even the associated distribution will be fine. To this end, I strongly suggest to exploit box plots.
Author Response:
We thank the reviewer for this valuable suggestion. We have now included the standard deviations of the results and provided box plots to illustrate the distribution of RSSI, SNR, and PDR measurements. These additions enhance the clarity and statistical depth of the results.
Comment 11:
Section 3. The obtained results should be compared with a theoretical model properly suiting the considered operation scenario.
Author Response:
We agree and have now included a comparison with the log-distance path loss model, which is appropriate for rural propagation analysis. This comparison allowed us to highlight the differences between theoretical expectations and our empirical results, and to explain them in terms of local terrain and environmental characteristics.
Comment 12:
Section 3.5. I did not get the utility of such a Section, since it actually shows nothing. For instance, a good idea could be to measure the current drawn of the developed sensor node, and possibly estimate the battery lifetime.
Author Response:
We revised Section 3.5 to better explain the autonomy discussion. Due to the manual transmission setup, we did not measure current consumption in this study. However, we clarified the methodological constraints and explained that the analysis of energy consumption and battery lifetime will be systematically addressed in our next study on IoT ecosystem deployment for rural irrigation. This was made explicit in the revised text.
Comment 13:
Lines 397-400. I cannot see the point in stating such a claim.
Author Response:
We thank the reviewer for this observation. The statement mentioned has been removed from the revised manuscript to avoid ambiguity.
Reviewer 3 Report
Comments and Suggestions for Authors
This work presented the experimental evaluation of LoRaWAN connectivity in rural Mozambique and found that distance and altitude significantly affect performance. Despite operational constraints, the network consistently achieved a performance deviation (PDR) above 89% and remained autonomous for over 24 hours. The findings can promote national digital inclusion policies.
Although by applying the following comments, it can be more enhanced.
- Contributions of this research should be clarified at the end of the introduction section.
- The quality of Figure 2 should be enhanced.
- The research design should be modified by adding the workflow and the algorithm of the proposed method.
- The comparison of this work with some recent literature should be specified and explained in the form of table at the end of result section.
- The conclusion and future work are too long. It should be restructured in 1 or 2 short paragraphs (250 words).
-
Author Response
Comment 1:
Contributions of this research should be clarified at the end of the introduction section.
Author Response:
We thank the reviewer for this suggestion. We have added a closing paragraph in the Introduction that clearly summarizes the main contributions of this study. The paragraph highlights (i) the novelty of conducting the first documented experimental evaluation of LoRaWAN in rural Mozambique, (ii) the empirical insights regarding the impact of distance and relative altitude on network performance, and (iii) the implications for IoT adoption in low-infrastructure environments.
Comment 2:
The quality of Figure 2 should be enhanced.
Author Response:
We have improved Figure 2 by increasing its resolution and ensuring that all labels, coordinates, and graphical elements are clearly legible.
Comment 3:
The research design should be modified by adding the workflow and the algorithm of the proposed method.
Author Response:
We have added a new figure illustrating the workflow of the experimental procedure, including the setup, measurement collection, and analysis steps. A brief description of the algorithm used for data handling and analysis was also included to improve transparency of the methodology.
Comment 4:
The comparison of this work with some recent literature should be specified and explained in the form of table at the end of result section.
Author Response:
We appreciate this suggestion. A new comparative table has been added at the end of the Results section. This table summarizes and contrasts our findings with recent works on LoRaWAN performance in rural and vegetated environments, highlighting both similarities and differences.
Comment 5:
The conclusion and future work are too long. It should be restructured in 1 or 2 short paragraphs (250 words).
Author Response:
We revised the Conclusion and Future Work section to make it more concise. It now consists of two short paragraphs (approximately 240 words), summarizing the key findings and outlining future research directions in a clearer and more compact format.
Reviewer 4 Report
Comments and Suggestions for Authors
The paper addresses an important and timely problem—evaluating the feasibility of LoRaWAN deployments in disconnected rural areas, specifically in Mozambique. The field experiments are well-motivated and provide valuable empirical insights for IoT connectivity in low-infrastructure environments. The results, highlighting the impact of terrain elevation and distance on connectivity, are relevant for both academic researchers and policymakers focused on digital inclusion. However, the manuscript would benefit from additional methodological rigor, expanded analysis, and clearer presentation in certain sections.
The paper emphasizes its novelty as the first documented empirical study of LoRaWAN in rural Mozambique. While this contextual contribution is clear, the technical novelty is somewhat limited. The authors should more explicitly state how their methodology or results differ from prior studies such as other chirp signal AFDM instead of LoRaWAN. [R1]"A Novel Angle-Delay-Doppler Estimation Scheme for AFDM-ISAC System in Mixed Near-Field and Far-Field Scenarios," in IEEE Internet of Things Journal, vol. 12, no. 13, pp. 22669-22682, 1 July1, 2025.
Consider clarifying whether the main contribution is empirical validation, methodological lessons for deployment, or policy-oriented recommendations.
The study is based on a single-node setup and four georeferenced points within 1 km. While useful as a proof-of-concept, the limited scale reduces generalizability. The authors should discuss to what extent these results can be extrapolated to larger deployments with multiple nodes and gateways.
The reliance on manual restarts for packet transmission is a serious limitation.
The statistical correlation analysis between RSSI, SNR, and PDR is valuable. However, regression or propagation models (e.g., log-distance path loss model) would strengthen the analysis and allow comparison with theoretical expectations.
Energy consumption is only superficially discussed. Reporting detailed current consumption (transmit, idle, sleep) would enhance replicability and provide a stronger basis for claims about autonomy.
The discussion of alternative technologies (NB-IoT, Starlink, etc.) is too qualitative. Including approximate cost comparisons or energy requirements would give more weight to the argument for LoRaWAN.
Author Response
Comment 1:
The paper emphasizes its novelty as the first documented empirical study of LoRaWAN in rural Mozambique. While this contextual contribution is clear, the technical novelty is somewhat limited. The authors should more explicitly state how their methodology or results differ from prior studies such as other chirp signal AFDM instead of LoRaWAN.
Author Response:
We thank the reviewer for this important observation. We revised the Introduction and the Related Works section to more explicitly clarify how our study differs from existing works. In particular, we emphasized that while prior studies often focus on LoRaWAN in suburban or controlled environments, our contribution lies in providing the first empirical validation in a rural African context with limited infrastructure. This perspective complements the broader body of work and provides practical insights for digital inclusion policies.
Comment 2:
Consider clarifying whether the main contribution is empirical validation, methodological lessons for deployment, or policy-oriented recommendations.
Author Response:
We agree that clarifying the focus of our contribution is essential. In the revised manuscript, the closing paragraph of the Introduction now explicitly states that the primary contribution is empirical validation of LoRaWAN performance in rural Mozambique. Secondary contributions include methodological lessons learned from operating in disconnected environments, and insights relevant to policy discussions on digital inclusion.
Comment 3:
The study is based on a single-node setup and four georeferenced points within 1 km. While useful as a proof-of-concept, the limited scale reduces generalizability. The authors should discuss to what extent these results can be extrapolated to larger deployments with multiple nodes and gateways.
Author Response:
We fully acknowledge this limitation. In the revised Discussion section, we added a paragraph addressing the scalability of our results. While the current findings cannot be directly generalized to large deployments, they provide a baseline for expectations in similar terrains. We also explained that our ongoing research involves multi-node and multi-gateway setups for IoT-based irrigation systems, which will expand the generalizability of the conclusions.
Comment 4:
The reliance on manual restarts for packet transmission is a serious limitation.
Author Response:
We agree and now explicitly mention this as a methodological limitation in Section 2. While manual restarts constrained the number of packets, they were necessary given the experimental conditions. We clarified that future tests will incorporate automated periodic transmission mechanisms to overcome this limitation.
Comment 5:
The statistical correlation analysis between RSSI, SNR, and PDR is valuable. However, regression or propagation models (e.g., log-distance path loss model) would strengthen the analysis and allow comparison with theoretical expectations.
Author Response:
We thank the reviewer for this constructive suggestion. We have now included a comparison with the log-distance path loss model in Section 3. This addition allowed us to contrast empirical and theoretical results, highlighting the specific impact of terrain and environmental factors in our test area.
Comment 6:
Energy consumption is only superficially discussed. Reporting detailed current consumption (transmit, idle, sleep) would enhance replicability and provide a stronger basis for claims about autonomy.
Author Response:
We acknowledge this point. Due to the limitations of the field setup, we did not measure current consumption in different operating states. However, we clarified this limitation in Section 3.5 and cited relevant works that report such measurements for similar devices. We also indicated that detailed energy profiling will be a key component of our upcoming multi-node IoT ecosystem study.
Comment 7:
The discussion of alternative technologies (NB-IoT, Starlink, etc.) is too qualitative. Including approximate cost comparisons or energy requirements would give more weight to the argument for LoRaWAN.
Author Response:
We agree with this observation and have expanded the discussion in Section 4. Approximate cost and energy requirement comparisons for LoRaWAN, NB-IoT, and satellite-based solutions (e.g., Starlink) were included, based on available literature. This addition provides a more quantitative foundation for our argument regarding the suitability of LoRaWAN in rural Mozambican contexts.
Round 2
Reviewer 1 Report
Comments and Suggestions for Authors
The authors have satisfactorily addressed the comments from my previous review.
Author Response
Dear Sir
Thanks a lot for the feedback and your important contributions to improving our manuscript
Reviewer 2 Report
Comments and Suggestions for Authors
The paper notably improved after its revision. I have a final comment, regarding one of the comments I raised in the previous round. While the Authors state they included box plots showing the distributions of the results, such plots seems to be missing in the manuscript. Please, add the plots.
Author Response
Reviwer Comment:
We sincerely thank the Reviewer for the positive assessment of the revised manuscript and for acknowledging the improvements made. We carefully considered the final comment regarding the box plots. In this revised version, we have included the requested box plots to illustrate the distribution of the results across the measurement points (now Figure 8). These visualizations complement the statistical tables and correlation plots already presented, thereby strengthening the robustness of the analysis.
Additionally, we improved the resolution and clarity of the figures, particularly Figures 2 and 3, to ensure that all illustrations are easily interpretable and meet publication standards.
We are confident that these adjustments address the Reviewer’s remaining concerns and further enhance the quality and readability of the manuscript.
Response:
We sincerely thank the Reviewer for the positive assessment of the revised manuscript and for acknowledging the improvements made. We carefully considered the final comment regarding the box plots. In this revised version, we have included the requested box plots to illustrate the distribution of the results across the measurement points (now Figure 8). These visualizations complement the statistical tables and correlation plots already presented, thereby strengthening the robustness of the analysis.
Additionally, we improved the resolution and clarity of the figures, particularly Figures 2 and 3, to ensure that all illustrations are easily interpretable and meet publication standards.
We are confident that these adjustments address the Reviewer’s remaining concerns and further enhance the quality and readability of the manuscript.
Reviewer 4 Report
Comments and Suggestions for Authors
I have no further comments and suggest accepting the current version.
Author Response

(The authors gave the same response as above.)
